# Correlation between Oral Health-Related Quality of Life and Objectively Measured Oral Health in People with Ehlers–Danlos Syndromes

**DOI:** 10.3390/ijerph17218243

**Published:** 2020-11-08

**Authors:** Ole Oelerich, Johannes Kleinheinz, Daniel R. Reissmann, Jeanette Köppe, Marcel Hanisch

**Affiliations:** 1Department of Cranio-Maxillofacial Surgery, Research Unit Rare Diseases with Orofacial Manifestations (RDOM), University Hospital Münster, D-48149 Münster, Germany; ole.oelerich@ukmuenster.de (O.O.); johannes.kleinheinz@ukmuenster.de (J.K.); 2Department of Prosthetic Dentistry, Center for Dental and Oral Medicine, University Medical Center Hamburg-Eppendorf, Martinistrasse 52, D-20251 Hamburg, Germany; d.reissmann@uke.de; 3Institute of Biostatistics and Clinical Research, University of Münster, D-48149 Münster, Germany; Jeanette.koeppe@ukmuenster.de

**Keywords:** rare diseases, Ehlers–Danlos syndromes, oral health, OHIP-14, EDS, patient-reported outcome, PhOX

## Abstract

It is a well-known fact that people with Ehlers–Danlos syndromes (EDS) report a worse oral health-related quality of life (OHRQoL) than the general population. The aim of this study was to examine whether there is a correlation between the subjective OHRQoL and the objectively measured oral health in people with EDS. To determine the subjective OHRQoL, the German version of the 14-item Oral Health Impact Profile (OHIP-14) questionnaire was used. Furthermore, all participants in the study were clinically examined, and the measured parameters were recorded using the Physical Oral Health Index (PhOX). Altogether, records of 46 participants were collected. The median (interquartile range (IQR)) of the OHIP-14 was 17 (23) points and the median of the PhOX was 73 (16) points. However, a statistically significant correlation could not be demonstrated (*r* = −0.240, *p* = 0.108). In the observed cohort, for participants who reported having pain often or very often, the OHIP score (median = 24, IQR = 18, *p* < 0.01) was higher than the score of the group given by participants who never or intermittently experienced pain (median = 8, IQR = 18). In conclusion, patients with EDS showed a reduced OHRQoL, although it was not possible to observe an association between the subjective OHRQoL and the objective oral health. However, participants who often or very often suffer from pain in their tooth, mouth, and jaw areas showed a reduced OHRQoL.

## 1. Introduction

Ehlers–Danlos Syndromes (EDS) are characterized as rare diseases. In the European Union, a disease is categorized as “rare” if less than one in 2000 people is affected [1]. Approximately 29 million people in the European Union are suffering from at least one of 5000–8000 different rare diseases. In Germany alone, there are probably more than four million people suffering from these diseases [2]. The EDS describe hereditary connective tissue diseases, which were originally subdivided into 11 subtypes according to the Berlin Nosology from 1988 [3]. With growing information about the origin of EDS, the nosology was revised in 1997 and the six-type distinctive Villefranche nosology was established [4]. Today, the syndromes can be classified into 13 subtypes according to the International Classification of Ehlers–Danlos Syndromes established in 2017 [5].

Almost 90% of persons suffering from EDS are affected by the classical EDS type and the hypermobile EDS type. The vascular type of EDS affects up to 10% of the patients. The periodontal type, which is particularly relevant in dentistry, occurs rather rarely [3,4,6,7]. The characteristic manifestations of the Ehlers–Danlos Syndromes are hypermobility of the joints, increased ductility of the skin, and weakness of the connective tissue. These symptoms can be attributed to flawed collagen synthesis or mutations [8]. The most frequently affected types are particularly type I, type III, and type V collagens (COL1A1, COL3A1, COL5A1). The affected genes encode for the fibrillary collagens and, through mutations, lead to common symptoms such as hypermobility of the joints, fragile skin, or even vascular damage [8]. The periodontal types lead to missense mutations in the C1R and C1S gene sections of Complement Factor 1 [9].

Since collagens are found in every connective tissue of the body, manifestations of EDS can also be seen in the teeth, mouth, and jaw areas. These oral and mandibular manifestations include thin and fragile mucosa, poor wound healing with increased tendency to bleeding, increased tooth mobility, and tooth and root form anomalies. Moreover, cranio-mandibular dysfunctions (CMD) occur more often [10]. The periodontal type is characterized by an early onset of severe periodontitis, which is accompanied by lack of attached gingiva and serious hematomas with bruises [5,11,12].

A study by Berglund et al. [13] identified four conditions “living with fear, living with pain, feeling stigmatized, and experiences of non-affirmation in healthcare”, which all lead to a reduced oral health-related quality of life (OHRQoL). The study shows just how many different aspects restrict the lives of patients with EDS in various ways, thus hindering a “normal life”. Bohner et al. [14] observed that patients affected by rare diseases and who report oral symptoms have an Oral Health Impact Profile (OHIP) score 6.45 points higher than patients without oral symptoms. Similarly, each year of delayed diagnosis has a negative impact on OHRQoL. The oral manifestations are, therefore, associated with a reduced oral health-related quality of life (OHRQoL) in patients with EDS.

Practitioners often lack knowledge about rare diseases, making it difficult for them to adequately treat patients suffering from a rare disease. This is why those patients are often dissatisfied with the healthcare system here in Germany [15]. Further studies are needed to examine how much practitioners know about rare diseases and what the German healthcare system must do to better the situation for those affected.

So far, however, it has only been demonstrated that patients with EDS are subjectively affected by a worse OHRQoL compared to the general populace. Because of the oral manifestations of EDS, the patients can also experience symptoms of impaired oral health. As a primary question of this study, we hypothesized that there is a significant correlation between objectively measured and subjectively reported oral health, i.e., H0: |*r*| = 0.

## 2. Materials and Methods

Data were collected over a period of 6 months (August 2019–January 2020). Furthermore, there was a positive vote of the Ethics Commission of the Chamber of Dentists of Westphalia-Lippe and the Westphalia Wilhelms University Münster (2019-404-f-S). The datasets supporting the conclusions of this article are available at the Department of Cranio-Maxillofacial Surgery, University Hospital Münster, Germany.

### 2.1. Participants

The participants had to be at least 18 years old and be affected by one of the Ehlers–Danlos syndromes. Patients who decided to participate in the context of the annual meeting of the “Ehlers Danlos Initiative e.V.” support group from 20–22 July 2019 in Bad Kissingen, Germany or after a visit to the special consultation on “Rare illnesses with an effect on oral health” were included in the study. Only those participants who filled out the Oral Health Impact Profile (OHIP-14) questionnaire for determining the OHRQoL and who participated in the clinical examination and data collection using the Physical Oral Health Index (PhOX) were included in the study.

### 2.2. Assesment of Physical Oral Health (PhOX)

The PhOX was developed for the purpose of recording and quantifying all aspects of the physical oral health of subjects. It consists of the self-assessed extraoral findings and intraoral findings. These categories are further subdivided into five subcategories with altogether 14 items (see Appendix A; Table 1).

Each criterion is evaluated on a five-point ordinal rating scale ranging from 0 to 4 and, depending on relevance, weighted either one, two, or three times (see Table 1). This yields a total score of 0–100 points, whereby 0 points is the worst possible and 100 points the best possible physical oral health. The periodontal status of the participants was ascertained with the help of Ramfjord’s Periodontal Disease Index (PDI) [16] on teeth 16, 21, 24, 36, 41, and 44.

### 2.3. Assesment of OHRQoL (OHIP)

To determine the subjective OHRQoL, the OHIP-14G was used (see Appendix A) [17]; this is the German short version of the OHIP-49. The OHIP-14G consists of 14 items assessing the frequency of pain, restrictions, social or physical stress, discomfort, and difficulties relating to social life. These items are rated on a scale from 0 = never to 4 = very often, resulting in a total score ranging from 0, no impact, to 56, very high impact of oral health on quality of life.

### 2.4. Statistical Methods

Assuming a two-sided significance level of 0.05 and a sample size of at least 40 participants, it was calculated that a Spearman correlation coefficient of |*r*| = 0.42 could be detected with a power of 81%. This was considered clinically relevant since the correlation of healthy subjects was known from a previous study [18]. For the primary hypothesis, a *p*-value of <0.05 was considered statistically significant. Further analyses were explorative (hypothesis generating) and not confirmatory, and, thus, interpreted accordingly. OHIP-14 score and PhOX score were analyzed descriptively. To compare both scores for participants with pain often or very often and with pain never or sometimes, a Mann–Whitney U-test was used. Statistical analyses were performed using IBM SPSS Statistics for Windows, Version 26.0, (IBM Corp., Armonk, NY, USA) and SAS software V9.4, (SAS Institute Inc., Cary, NC, USA).

## 3. Results

### 3.1. Age and Diagnosis Period

A total of 46 participants were included in the study with 91.3% being female (see Table 2), i.e., four participants were male and 42 were female. The median age (interquartile range (IQR)) of the participants was 39.5 (23) years. At the time of diagnosis, the median age (IQR) was 34 (25) years. The participants had to wait 20 (27) years for their diagnosis after the presence of the first symptoms.

### 3.2. EDS Diagnoses

The contribution of different EDS subtypes can be seen in Table 3. Eleven participants (23.9%) did not give any details about their subtype.

### 3.3. OHRQoL

The median of the OHIP-14 scores was 17 (IQR = 23, range 0–42; see Figure 1). A total of 22 participants stated that their OHRQoL was sometimes or even often impaired (see Table 4 for all items). A total of 25 participants stated that they often had pain in the mouth area. Twelve of them even stated that the pains occurred very often. Apart from pain, the participants also suffered from anxiety (32.6% often or very often) and had difficulty relaxing (34.8% often or very often). The problems occurred in various everyday situations, and their diet was, thus, also affected. This is mainly due to the fact that 32.6% of the participants stated that they considered certain foods unpleasant and less so because participants had to interrupt their meals or stated that their diet was unsatisfactory. Furthermore, 12 participants stated that they often felt unsure with respect to their teeth, dentures, or mouth. Eight participants stated that this feeling of insecurity occurred very often.

### 3.4. Physical Oral Health

The median PhOX score was 73 (IQR = 16, range 38–91; see Figure 1). All participants had at least 14 teeth; 24 participants had a full set of teeth (3rd molar excluded). A total of 20 participants had gingival pockets of 3.5–5.5 mm on 1–3 teeth (all items can be found in Table 5). Thirteen participants had gingival pockets of 3.5–5.5 mm on more than three teeth or gingival pockets of more than 5.5 mm on 1–3 teeth. On palpation of the salivary glands, the chewing muscles, and the mandibular joints, 18 participants felt moderate unfamiliar pain or slight familiar pain. Eight participants felt severe familiar pain. Only 11 participants did not feel any pain on palpation. The study from Reissmann et al. [18] suggests that moderate unfamiliar pain or slight familiar pain should be treated equally and, therefore, should have the same impact on the score. For 32 of the participants, the incisal edge distance with a maximum mouth opening was over 40 mm. Only four participants had a maximum incisal edge distance of less than 30 mm with more than 5 mm of passive ductility. Ten participants stated that they experienced unpleasant sensations several times a week, including a burning sensation, hypersensitivity, or numbness in the mouth area. Twelve participants stated that they experienced these unpleasant sensations daily.

### 3.5. Relationship between OHRQoL and Physical Oral Health

In the observed cohort, a significant correlation between OHIP-14 and PhOX scores could not be proven (*r* = −0.24, *p* = 0.108). Figure 2 shows the association between OHIP-14 and PhOX scores (see Appendix A, for a full overview including individual values of age, sex, and OHIP and PhOX score for each participant). In the observed cohort, the OHIP-14 score was worse in participants who experienced oral pain often or very often (29 participants; 63.0%) than in the group who never or rarely had pain (*p* < 0.01). However, no difference could be observed for the PhOX score (*p* > 0.05). The median (IQR) OHIP-14 score for participants who often or very often had pain was 24 (18) (see Figure 3). On the other hand, for the group having pain never to sometimes, a median (IQR) score of 8 (11) was found. The median (IQR) PhOX score for participants who often or very often have pain was 70 (15), compared to the group with having pain never to sometimes (median = 74, IQR = 13; see Figure 3).

## 4. Discussion

The aim of this study was to find out if this subjective perception of the OHRQoL correlates with objectively measured oral health. The evaluated OHIP 14 scores of this study (median 17, IQR 23, range 0–42) are comparable with the findings of similar studies on OHRQoL for patients with Ehlers–Danlos Syndromes (Hanisch et al. 2018 [15]; Berglund et al. 2012 [19]). A study by Reissmann et al. [18] yielded a correlation (*r* = 0.41) between the OHIP-14 and PhOX by analyzing 609 dental patients, because of which we hypothesized that there is a significant correlation between the OHRQoL and the objectively measured oral health in our cohort. 

However, patients with EDS sometimes report problems and restrictions even with good oral health, even if these problems cannot be objectively understood from a dental perspective. The results showed that the objectively measured oral health of participants with EDS often appears good, even if the participants suffer from a low OHRQoL.

In our cohort, pain was a significant factor worsening the OHRQoL, as seen in Table 4. Since more than 50% of the participants had pain often to very often, this item had the single biggest negative impact on the OHIP-14 score. This is comparable to similar findings in the Berglund study, in which pain was also a significant factor with a negative influence on the OHRQoL.

We observed that the OHIP-14 score for participants who often or very often experienced pain in the mouth area had higher scores compared to participants who never or only rarely felt pain (*p* < 0.01), without showing a difference in the PhOX score (*p* > 0.05). This discrepancy may possibly complicate the care of patients with EDS in daily clinical practice and increase the level of distress for those patients, as the causes of pain cannot be identified from a dental perspective.

The problem that pain cannot necessarily be clinically verified was described previously by patients with EDS. In 1977, Sancheti et al. demonstrated that chronic pain is a manifestation of Ehlers–Danlos Syndromes [20]. In future studies, the possible association between chronic pain and reduced OHRQoL should be addressed. 

In addition to chronically manifested pain, patients with EDS often suffer from craniomandibular dysfunction [10]. In our cohort, only 11 participants (23.9%) did not feel any pain on palpation of the salivary glands, chewing muscles, or mandibular joint. Pain on palpation, especially of the chewing muscles and mandibular joints, is a common symptom in the diagnosis of CMD [21]. Previous studies have shown that patients with EDS have a significant higher prevalence of CMD symptoms compared to a control group [22]. In addition to affecting the temporomandibular joint, patients with EDS often show an increased mouth opening (for 69.5% of our cohort the maximal incisal edge was above 40mm), having a higher risk of sudden meniscal and/or condylar jaw dislocation, which can lead to inflammation of the affected regions [23].

To help those patients who suffer from CMD and to increase their OHRQoL, CMD-specific physiotherapy is recommended to counteract damage to the temporomandibular joint (TMJ) [23]. This therapy can involve both changes in body posture and changes in everyday life, such as chewing habits or stress management techniques. Mitakides et al. even recommended this therapy preventively for all patients with EDS [23]. They also pointed out that the treatment of EDS patients can be quite challenging even for therapists experienced in handling CMD and recommended treatment by specialists who are well versed with EDS symptoms [10].

Another well-known problem in patients with rare diseases is the long period between the first appearance of symptoms and the diagnosis. On average, it takes 7 years until patients with rare diseases are finally diagnosed. This long time period is accompanied by a lot of stress and uncertainty for the affected patients [24]. With a median of 20 years, the period in this study was considerably higher, negatively impacting the patients and their OHRQoL even more. Due to the long diagnosis period, the pain with which patients have to live is further increased and can manifest itself chronically [25].

On one hand, this presents the therapist with a complex task of providing adequate care for the patients. On the other hand, patients with EDS can quickly become dissatisfied. The study by Hanisch et al. (2018) showed that the patients were more satisfied with the therapist, but rather perceived the support provided by the German health system as insufficient [15]. One aim should be to train the therapists in such a way that they can identify an EDS patient as such and treat them adequately or to point out specific points of contact.

Practitioners are faced with the difficult task of treating patients adequately, even if the patient is not showing corresponding clinical symptoms. The aim should be to assess the OHRQoL of each patient independent of the objectively measured oral health and treat the patient accordingly. To achieve this and to better the OHRQoL in patients with EDS in general, the awareness of rare diseases such as EDS must increase, shortening the time period of diagnosis and relieving patients of their restricted lives.

### Limitations

Both the OHIP-14 and the PhOX are validated scores which have previously been used in studies. The participation in the study revealed an enormous imbalance between the sexes. Out of the 46 participants, only four were male. This could be explained by the fact that women are more often involved in support groups as observed in a similar study by Hanisch et al. [26]. However, a previous study showed that there was no connection with sex for measuring OHRQoL in patients with rare diseases [14]. Therefore, it is unlikely that the imbalance between sexes has an impact on the outcome of the study. Further studies with higher numbers of participants would be desirable, but high numbers of participants are often difficult to realize, especially in people with rare diseases.

## 5. Conclusions

Patients with Ehlers–Danlos Syndromes showed worse OHRQoL; however, no significant correlation between OHRQoL and physical oral health could be proven. The hypothesis that an objectively good oral health is also associated with a good OHRQoL does not seem to apply to patients with Ehlers–Danlos syndromes. Patients who often or very often suffer from pain in the tooth, mouth, and jaw area were especially observed to have a reduced OHRQoL.

The PhOX can be used to assess the objectively measured oral health in patients with EDS accordingly, but the OHRQoL should be individually assessed for each patient independent of their objectively measured oral health.

## Figures and Tables

**Figure 1 ijerph-17-08243-f001:**
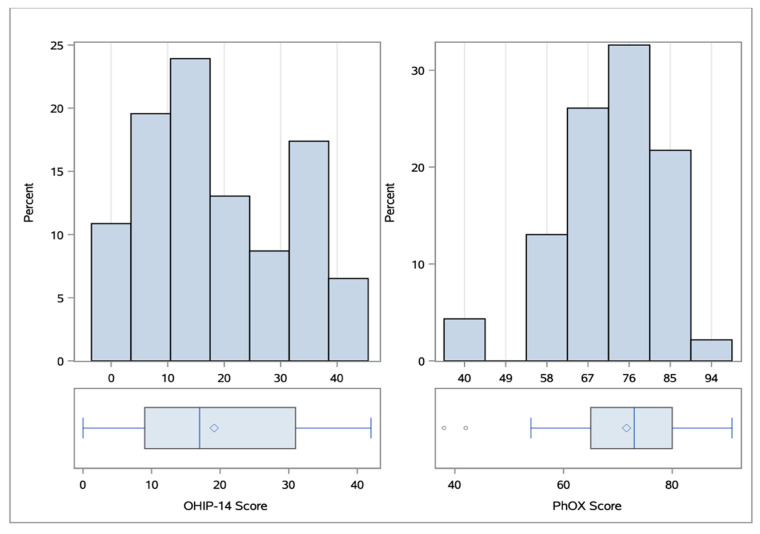
Distribution of the values of 14-item Oral Health Impact Profile (OHIP-14) and PhOX with related boxplots. The median (IQR) of the OHIP-14 score was 17 (23) and the median of the PhOX was 73 (16).

**Figure 2 ijerph-17-08243-f002:**
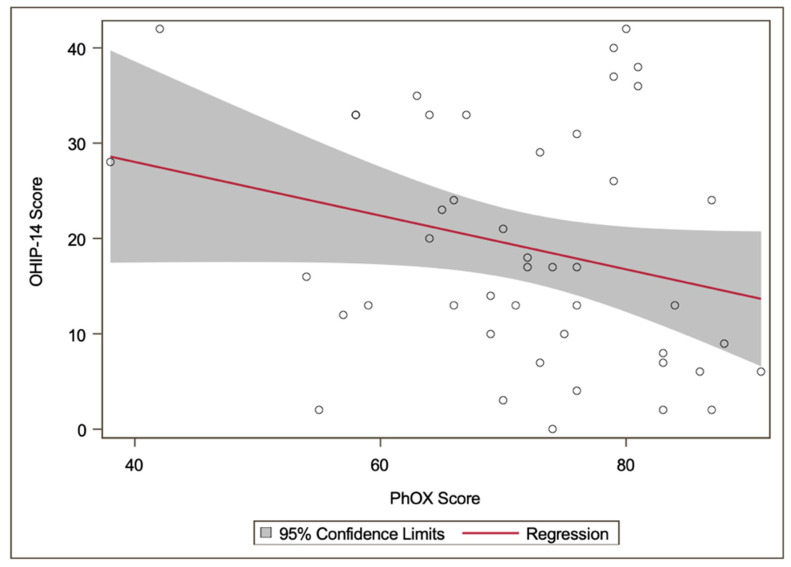
Scatterplot of OHIP-14 and PhOX scores with linear regression (Spearman correlation coefficient *r* = −0.24) and 95% confidence limits of the regression coefficient.

**Figure 3 ijerph-17-08243-f003:**
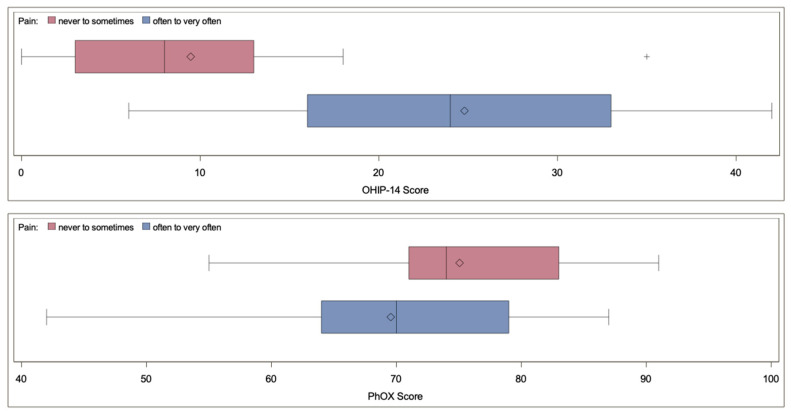
Boxplots for OHIP-14 score and PhOX score categorized into two groups. The red group experienced pain never to sometimes, where the blue group experienced pain often to very often.

**Table 1 ijerph-17-08243-t001:** Physical Oral Health Index (PhOX) domains and weights with the range for the values of each item. Translated domains and items from the Physical Oral Health Index.

Domain	#	Item	Weight	Range
Teeth and surrounding tissue	1	Number Of Teeth	3	0–12
	2	Tooth Structure	3	0–12
	3	Periodontium	3	0–12
	4	Endodontia	2	0–8
Soft tissue intraoral	5	Surface	1	0–4
	6	Color	2	0–8
	7	Moisturization	1	0–4
Soft tissue and jaw	8	Pain On Palpation	1	0–4
	9	Continuity	1	0–4
	10	Proportion	1	0–4
Function	11	Mouth Opening	1	0–4
	12	Supporting Area	3	0–12
Perception	13	Pain	2	0–8
	14	Paresthesia	1	0–4

**Table 2 ijerph-17-08243-t002:** Patient-specific information regarding age and diagnosis. IQR, interquartile range.

	Minimum	Maximum	Median	IQR
Age ^1^	19	82	39.5	23
Time of diagnosis ^1^	0	79	34	25
Time between first ^1^ symptoms and diagnosis	0.12	60	20	27

^1^ In years.

**Table 3 ijerph-17-08243-t003:** Contribution of the subtypes of Ehlers–Danlos syndromes (EDS) in the study cohort.

# ^2^	Clinical EDS Subtype	Abbreviation	*N* (%)
1	Classical EDS	cEDS	6 (13.0)
2	Classic-like EDS	clEDS	1 (2.2)
4	Vascular EDS	vEDS	2 (4.3)
5	Hypermobile EDS	hEDS	24 (52.0)
9	Brittle cornea syndrome	BCS	2 (4.3)
-	Not provided by patient	-	11 (23.9)

^2^ Subtype number according to the International Classification of Ehlers–Danlos Syndromes established in 2017 [5].

**Table 4 ijerph-17-08243-t004:** Distribution of responses of individual OHIP-14 items.

OHIP-14	0 (%)	1 (%)	2 (%)	3 (%)	4 (%)
Have you had any problems with your teeth, mouth or dentures in the past month…					
…making it difficult to pronounce certain words?	30 (65.2)	5 (10.9)	5 (10.9)	4 (8.7)	2 (4.3)
…that made you feel that your sense of taste was impaired?	26 (56.5)	8 (17.4)	7 (15.2)	3 (6.5)	2 (4.3)
…that gave you the impression that your life in general was less satisfying?	15 (32.6)	6 (13.0)	12 (26.1)	10 (21.7)	3 (6.5)
…making it difficult to relax?	6 (13.0)	14 (30.4)	10 (21.7)	8 (17.4)	8 (17.4)
Has it often happened in the past month due to problems with your teeth, mouth or dentures…					
…that you felt tense?	8 (17.4)	10 (21.7)	13 (28.3)	10 (21.7)	5 (10.9)
…that you had to interrupt meals?	19 (41.3)	9 (19.6)	13 (28.3)	3 (6.5)	2 (4.3)
…that you were uncomfortable eating certain foods?	11 (23.9)	7 (15.2)	13 (28.3)	7 (15.2)	8 (17.4)
…that you have been a bit irritable with other people?	17 (37.0)	13 (28.3)	11 (23.9)	5 (10.9)	0 (0)
…that you had difficulty doing your usual jobs?	18 (39.1)	9 (19.6)	8 (17.4)	7 (15.2)	4 (8.7)
…that you were completely unable to do anything?	26 (56.5)	10 (21.7)	7 (15.2)	1 (2.2)	2 (4.3)
…that you have been a bit embarrassed?	25 (54.3)	12 (26.1)	5 (10.9)	4 (8.7)	0 (0)
…that your diet has been unsatisfactory?	26 (56.5)	9 (19.6)	5 (10.9)	4 (8.7)	2 (4.2)
In the past month, did you have…					
…any pain in your mouth?	6 (13.0)	1 (2.2)	14 (30.4)	13 (28.3)	12 (26.1)
…a feeling of uncertainty about your teeth, mouth, or dentures?	12 (26.1)	8 (17.4)	8 (17.4)	12 (26.1)	6 (13.0)

Responses on Likert scale: 0—never, 1—hardly ever, 2—occasionally, 3—fairly often, 4—very often. OHIP-14—translated items of the German short form of the Oral Health Impact Profile (OHIP-14G).

**Table 5 ijerph-17-08243-t005:** Distribution of the individual 14 PhOX items among the participants.

PhOX Item	0 (%)	1 (%)	2 (%)	3 (%)	4 (%)
Number of teeth	0 (0)	0 (0)	4 (8.7)	18 (39.1)	24 (52.2)
Tooth structure	1 (2.2)	21 (45.7)	5 (10.9)	11 (23.9)	8 (17.4)
Periodontium	0 (0)	13 (28.3)	20 (43.5)	4 (8.7)	9 (19.6)
Endodontia	0 (0)	1 (2.2)	2 (4.3)	19 (41.3)	24 (52.2)
Surface	1 (2.2)	1 (2.2)	4 (8.7)	19 (41.3)	21 (45.7)
Color	0 (0)	15 (32.6)	0 (0)	10 (21.7)	21 (45.7)
Moisturization	0 (0)	3 (6.5)	5 (10.9)	0 (0)	38 (82.6)
Pain on palpation	8 (17.4)	7 (15.2)	18 (39.1)	2 (4.3)	11 (23.9)
Continuity	0 (0)	0 (0)	0 (0)	0 (0)	46 (100.0)
Proportion	0 (0)	0 (0)	1 (2.2)	5 (10.9)	40 (87.0)
Mouth opening	0 (0)	4 (8.7)	10 (21.7)	10 (21.7)	22 (47.8)
Supporting area	0 (0)	1 (2.2)	2 (4.3)	2 (4.3)	41 (89.1)
Pain	13 (28.3)	16 (34.8)	11 (23.9)	2 (4.3)	4 (8.7)
Paresthesia	10 (21.7)	8 (17.4)	6 (13.0)	10 (21.7)	12 (26.1)

Evaluated on a five-point ordinal scale with 0 being the worst and 4 being the best possible result.

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
