# Peer review of "Correlation between Oral Health-Related Quality of Life and Objectively Measured Oral Health in People with Ehlers–Danlos Syndromes"

_ijerph, 2020, doi:10.3390/ijerph17218243_

Round 1
Reviewer 1 Report
This is a study on Oral Health in People 4 with Ehlers-Danlos Syndromes. This is a study to examine the relationship between objective and subjective oral health of subjects with special diseases other than general subjects.
- There is no correspondence between the purpose and title.
- General information on participants is lacking. For example, please provide a table with the most common items, including mean and gender.
- Does the title of Physical Measured Oral Health mean objectively measured oral health? If so, the English expression needs to be modified.
- Line on 74-75, what does physical measured OHRQoL mean?
- Are Physical Oral Health Index (PhOX) and physical measured OHRQoL used interchangeably?
- What does Figure 1 mean? You must present the correct values on the bar graph.
- In Figure 2, you need to provide a regression value.
- Figure 3 has no values. What does this mean?
- Minors
- Check and correct the notation form of References in text
- Correction of uppercase and lowercase letters in English
Author Response
We would like to thank the editor and the reviewers for their time spent on reviewing our manuscript and their helpful comments. Their suggestions have been implemented in the manuscript. In this letter, we respond point-by-point to the comments and explain the revisions.
All changes to the manuscript were highlighted using the "Track Changes" function in Microsoft Word.
We hope the manuscript is now suitable for publication in the International Journal of Environmental Research and Public Health.
Reviewer 1:
Comments and Suggestions for Authors
This is a study on Oral Health in People 4 with Ehlers-Danlos Syndromes. This is a study to examine the relationship between objective and subjective oral health of subjects with special diseases other than general subjects.
- There is no correspondence between the purpose and title.
The title was changed to better fit the purpose of the article.
- General information on participants is lacking. For example, please provide a table with the most common items, including mean and gender.
Thank you for noticing, supplement S3 was added for a full overview over age, sex, OHIP- and PhOX-Score for each participant. Throughout the whole article we only used median and not mean because it offers a more realistic representation of the small cohort size.
- Does the title of Physical Measured Oral Health mean objectively measured oral health? If so, the English expression needs to be modified.
The title was changed for a more accurate description of the study. We only used the expression “Physical Measured Oral Health” to lay the focus on the Physical Oral Health Index used. The new title gives a better representation of the aim of the study.
- Line on 74-75, what does physical measured OHRQoL mean?
This expression must have been left in by mistake and there is nothing like the physical measured OHRQoL. I changed the expression to objectively measured oral health (line 91-92). Thank you for noticing.
- Are Physical Oral Health Index (PhOX) and physical measured OHRQoL used interchangeably?
See bullet point 4
- What does Figure 1 mean? You must present the correct values on the bar graph.
There must have been a problem with exporting the Figures as PDFs as they did not show all the information they should have. I fixed all the figures by implementing them using TIFF format. Now they should all show the correct values.
- In Figure 2, you need to provide a regression value.
I added the regression value to the figure description, thank you (line 191-192).
- Figure 3 has no values. What does this mean?
See bullet point 6.
- Minors
- Check and correct the notation form of References in text
- Correction of uppercase and lowercase letters in English
checked
Reviewer2:
Comments and Suggestions for Authors
In this paper, oral health-related quality of life (OHRQoL) is correlated to clinical findings. However, the clinical findings are based the use of a questionnaire published in a paper in German with no proper reference:
Reißmann DR. Eine neue und innovative Methode zur umfassenden Beschreibung der physischen 282
Mundgesundheit: Der Physical Oral Health Index (PhOX) n.d.:
There must have been an export problem with Zotero, thank you for noticing. I checked and fixed the reference. The correct reference is:
Reissmann DR, Aarabi G, von Wolff A, et al. The Physical Oral Health Index: reliability and validity. J Dent Res 2015;94(Special Issue A):1465 (line 360-361)
The conclusion is that there is no significant correlation. The obvious question is "Was the appropriate clinical parameters used"? To me it seems like there was no correlation between the OHIP-14 and the physical health questionnaire used, while certain clinical parameters indeed showed such a correlation.
Interesting point. We still think that there is a correlation between the PhOX and the OHIP-14, but this correlation was simply missing because of the complexity of problems in our cohort. As the previous study by Reissmann et al. revealed, in a general population there was indeed a correlation of r=0.41. This leads to the conclusion that the Physical Oral Health Index is a proper tool to accurately describe the patients objectively measured oral health in consideration of the OHRQoL. This, however, doesn´t seem to be the case in our cohort. Which leads to the conclusion that patients with EDS can be examined using the PhOX to analyze their objectively measured oral health, but their OHRQoL should be examined individually and taken into account regardless of the given PhOX score to treat the patients adequately.
As stated in the conclusion: Particularly, participants who often or very often suffer from pain in their tooth, mouth and jaw areas showed a reduced OHRQoL.
We highlighted the factor of pain, because the patients who often or very often suffer from pain in the given region not only showed a reduced OHRQoL but also a reduced objectively measured oral health.
Therefore, I suggest that the paper is rewritten based on these findings, downsizing the importance of the correlation between (OHRQoL) and the questionnaire used. One conclusion is that the questionnaire used does not seem to appropriately evaluate oral health in patients with EDS.
Thank you for all the annotations. The discussion was rewritten, increasing the importance of the OHRQoL in patients with EDS. We concluded that the questionnaire used does not seem to appropriately evaluate oral health regarding OHRQoL in patients with EDS. The Physical Oral Health Index can be used to assess the objectively measured oral health but should not be used to draw conclusions regarding the general oral health which also included OHRQoL. The OHRQoL should be assessed and evaluated separately regarding patients with EDS.
Reviewer 3:
Comments and Suggestions for Authors
The aim of the study seems disconnected from the introduction. More could be added about the importance of OHRQoL and the importance of the research.
Thank you for the annotations. The introduction was rewritten, adding more about the importance of OHRQoL and the importance of research.
The use of information in parentheses make the introduction difficult to understand. Perhaps use bullet points for the types of EDS and the previous names.
We gathered the information in parentheses in table 3 to make it easier to understand and also offering information regarding the distribution of subtypes all in one table.
Perhaps use more shorter paragraphs to help clarify main ideas. The aim of the study is referred to later in the paper as the "hypothesis" but is not written in the style of hypothesis.
The aim of the study was corrected and rewritten in the style of hypothesis (line 86-87). Thank you for noticing.
In the methods, the authors mention that the PhOX tool contains a portion that is self-assessment. How was this information used in the analysis? IF comparing subjective OHRQoL to clinically determined oral health status, should the self-assessment section be excluded from analysis? Or perhaps look at it separately in comparison to OHRQoL.
We added table 1 to better explain each domain with corresponding items. The portion of self-assessment within the PhOX assess pain and paresthesia. Those items were included in the analysis because they were also used in the previous study by Reissmann et al. They are also essential to correctly evaluate the PhOX. The self-assessment question regarding pain was assessed using the same 5-point ordinal scale used in the OHIP-14.
More information about the items included in the measurement tools should be included in the methods and also reported in the results. What types of factors lead to the higher or lower scores? A table could be added highlighting these results.
Thank you for the annotation. Tables were added to show what types of factors lead to a higher or lower score (table 1). We also added table 4 and 5 to give detailed information regarding each item of the OHIP-14 and PhOX score.
The text of the results for EDS diagnosis, physical oral health, and OHRQoL should be simplified, and more complete information should be presented in table. More explanation of the unfamiliar pain vs. familiar pain measured by the PhOX is required. What are the implications of this distinction?
The text was simplified adding table 2,3,4 and 5 to offer more complete information. The explanation regarding unfamiliar vs familiar pain was added (line 173-175). The implications are, that from a practitioner’s view, moderate unfamiliar and slight familiar pain should be treated equally and should be given the same score.
Figure 2 is not referenced in the text.
Thank you for noticing, the figure and table references were checked and corrected.
The term "control group" is used incorrectly in section 3.5.
Thank you for noticing, the term “control group” was an expression error. We changed the expression accordingly.
You are stratifying by pain status. There is no real control group. What is the importance of this stratification? Is it not obvious that those with pain would have poorer OHRQoL? If the importance is that PhOX is similar even though pain AND OHRQoL varies - it is not made clear in these results.
The aim was to see, which factor has the most negative impact on the OHRQoL in our cohort. Table 4 shows that question 13, regarding pain, was indeed this factor with a median score of 3. Pain is a well-known factor worsening the OHRQoL in patients with EDS, that is why we wanted to make this distinction here. The results were rewritten, highlighting the importance of pain even though no difference was observed in the PhOX.
The paragraph on Page 5, rows 156-160 belongs in the discussion section.
Thank you for noticing, the given paragraph was moved to the discussion section (line 205-206).
In the discussion section, the first sentence says, "subjective perception of the OHRQoL can also be confirmed by clinical examination." "Confirmed" is not the correct word, as by definition the OHRQoL is a personal opinion or feeling that cannot be confirmed. Restate this aim differently. The main point and hypothesis generating results are difficult to understand in the discussion, separate these ideas into different paragraphs with clearer explanation.
The aim was restated in line 202-203. We also generated a clearer results section with more paragraphs containing the different ideas.
Page 6 paragraph between lines 189-199 is reporting results about anxiety and difficulty relaxing. These should be moved to results section and reported in greater detail.
The given paragraph was moved to the results section (line 154-155) and was further reported and visualized in table 4.
The discussion of treatment of pain seems irrelevant. Thus, it needs a stronger link to the results. Page 7 line 209, "less dissatisfied" is unclear. Should it be "more satisfied?"
This falsely expression was changed, thank you (line 258). Also, the discussion of treatment of pain was rewritten.
What reference or evidence do you have for stating that patient's symptoms are ignored when not verified? (page 7, line 212-213.
The give phrase was rewritten (line 263-168). We tried to emphasize that practitioners should treat patients with EDS differently because of the missing correlation between the OHRQoL and the objectively measured oral health. We also tried to emphasize the importance of information regarding EDS, making it more known, thus shortening the period of time patients usually wait between the first symptoms and the final diagnosis.
Add CMD to list of abbreviations.
Thank you for noticing, CMD and IQR were added to the list of abbreviations (line 295-297).
Reviewer 2 Report
In this paper, oral health-related quality of life (OHRQoL) is correlated to clinical findings. However, the clinical findings are based the use of a questionnaire published rin a paper in German with no proper reference:
Reißmann DR. Eine neue und innovative Methode zur umfassenden Beschreibung der physischen 282
Mundgesundheit: Der Physical Oral Health Index (PhOX) n.d.:
The conclusion is that there is no significant correlation. The obvious question is "Was the appropriate clinical parameters used"? To me it seems like there was no correlation between the OHIP-14 and the physical health questionnaire used, while certain clinical parameters indeed showed such a correlation.
As stated in the conclusion: Particularly, participants who often or very often suffer from pain in their tooth, mouth and jaw areas showed a reduced OHRQoL.
Therefore I suggest that the paper is rewritten based on these findings, downsizing the importance of the correlation between (OHRQoL) and the questionnaire used. One conclusion is that the questionnaire used does not seem to appropriately evaluate oral health in patients with EDS.
Author Response

(The authors gave the same response as above.)

Reviewer 3 Report
The aim of the study seems disconnected from the introduction. More could be added about the importance of OHRQoL and the importance of the research. The use of information in parentheses make the introduction difficult to understand. Perhaps use bullet points for the types of EDS and the previous names. Perhaps use more shorter paragraphs to help clarify main ideas. The aim of the study is referred to later in the paper as the "hypothesis" but is not written in the style of hypothesis.
In the methods, the authors mention that the PhOX tool contains a portion that is self assessment. How was this information used in the analysis? IF comparing subjective OHRQoL to clinically determined oral health status, should the self assessment section be excluded from analysis? Or perhaps look at it separately in comparison to OHRQoL. More information about the items included in the measurement tools should be included in the methods and also reported in the results. What types of factors lead to the higher or lower scores. A table could be added highlighting these results.
The text of the results for EDS diagnosis, physical oral health, and OHRQoL should be simplified, and more complete information should be presented in table. More explanation of the unfamiliar pain vs. familiar pain measured by the PhOX is required. What are the implications of this distinction?
Figure 2 is not referenced in the text.
The term "control group" is used incorrectly in section 3.5. You are stratifying by pain status. There is no real control group. What is the importance of this stratification. Is it not obvious that those with pain would have poorer OHRQoL? If the importance is that PhOX is similar even though pain AND OHRQoL varies - it is not made clear in these results.
The paragraph on Page 5, rows 156-160 belongs in the discussion section.
In the discussion section, the first sentence says "subjective perception of the OHRQoL can also be confirmed by clinical examination." "Confirmed" is not the correct word, as by definition the OHRQoL is a personal opinion or feeling that can not be confirmed. Restate this aim differently. The main point and hypothesis generating results are difficult to understand in the discussion, separate these ideas into different paragraphs with clearer explanation.
Page 6 paragraph between lines 189-199 is reporting results about anxiety and difficulty relaxing. These should be moved to results section and reported in greater detail.
The discussion of treatment of pain seems irrelevant. Thus it needs a stronger link to the results. Page 7 line 209, "less dissatisfied" is unclear. Should it be "more satisfied?" What reference or evidence do you have for stating that patient's symptoms are ignored when not verified? (page 7, line 212-213.
Add CMD to list of abbreviations.
Author Response

(The authors gave the same response as above.)

Round 2
Reviewer 1 Report
I found many mistakes were corrected.
The authors supplemented figures and tables to present results in more clarity.